Cannabidivarin (CBDV) suppresses pentylenetetrazole (PTZ)-induced increases in epilepsy-related gene expression

Amada Naoki 1 2 3 Amada.Naoki@otsuka.jp n.amada@pgr.reading.ac.uk
Yamasaki Yuki 1 2 3
Williams Claire M. 2
Whalley Benjamin J. 1
1 School of Chemistry, Food and Nutritional Sciences and Pharmacy, The University of Reading , Reading, Berkshire , UK
2 School of Psychology and Clinical Language Sciences, The University of Reading , Reading, Berkshire , UK
3 Qs’ Research Institute, Otsuka Pharmaceutical, Co. Ltd. , Kagasuno, Kawauchi-cho, Tokushima , Japan
Benigni Ariela
Electronic publication date: 2013 Nov 21
Publication date: 2013
Volume: 1
Electronic Location ID: e214
Received 2013 Jun 18; Accepted 2013 Oct 30
Copyright: © 2013 Amada et al.
Copyright year: 2013
Copyright holder: Amada et al.
License: This is an open access article distributed under the terms of the Creative Commons Attribution License, which permits unrestricted use, distribution, and reproduction in any medium, provided the original author and source are credited.
License URL: https://creativecommons.org/licenses/by/3.0/

Keywords: Cannabinoid, Pentylenetetrazole, Cannabidivarin, qPCR, Seizure, Epilepsy

Funding: GW Pharmaceuticals Plc Otsuka Pharmaceutical Co, Ltd This work was supported by GW Pharmaceuticals Plc. and Otsuka Pharmaceutical Co, Ltd. The funders gave final approval to submit the prepared manuscript to peer review for publication.

==============================
To date, anticonvulsant effects of the plant cannabinoid, cannabidivarin (CBDV), have been reported in several animal models of seizure. However, these behaviourally observed anticonvulsant effects have not been confirmed at the molecular level. To examine changes to epilepsy-related gene expression following chemical convulsant treatment and their subsequent control by phytocannabinoid administration, we behaviourally evaluated effects of CBDV (400 mg/kg, p.o.) on acute, pentylenetetrazole (PTZ: 95 mg/kg, i.p.)-induced seizures, quantified expression levels of several epilepsy-related genes (Fos, Casp 3, Ccl3, Ccl4, Npy, Arc, Penk, Camk2a, Bdnf and Egr1) by qPCR using hippocampal, neocortical and prefrontal cortical tissue samples before examining correlations between expression changes and seizure severity. PTZ treatment alone produced generalised seizures (median: 5.00) and significantly increased expression of Fos, Egr1, Arc, Ccl4 and Bdnf. Consistent with previous findings, CBDV significantly decreased PTZ-induced seizure severity (median: 3.25) and increased latency to the first sign of seizure. Furthermore, there were correlations between reductions of seizure severity and mRNA expression of Fos, Egr1, Arc, Ccl4 and Bdnf in the majority of brain regions in the CBDV+PTZ treated group. When CBDV treated animals were grouped into CBDV responders (criterion: seizure severity ≤3.25) and non-responders (criterion: seizure severity >3.25), PTZ-induced increases of Fos, Egr1, Arc, Ccl4 and Bdnf expression were suppressed in CBDV responders. These results provide the first molecular confirmation of behaviourally observed effects of the non-psychoactive, anticonvulsant cannabinoid, CBDV, upon chemically-induced seizures and serve to underscore its suitability for clinical development.

Introduction

Epilepsy affects ∼1% of individuals and is often characterized by recurrent seizures. Many treatments are available but more effective and better-tolerated antiepileptic drugs (AEDs) with new mechanisms of actions are needed due to drug resistance (∼35%) and poor AED side-effect profiles (Kwan & Brodie, 2007).

Several cannabinoids (Δ9-tetrahydrocannabinol: Δ9-THC, cannabidiol: CBD, Δ9-tetrahydrocannabivarin: Δ9-THCV and cannabidivarin: CBDV) are anticonvulsant in a variety of animal models of seizure and epilepsy (Consroe & Wolkin, 1977; Hill et al., 2012a; Hill et al., 2010; Jones et al., 2010). Whilst CB1 cannabinoid receptor (CB1R) agonism is anti-epileptiform and anticonvulsant (Chesher & Jackson, 1974; Deshpande et al., 2007b; Wallace et al., 2003; Wallace et al., 2001), the notable psychoactivity associated with CB1R activation hinders the prospective clinical utility of this target. However, many plant cannabinoids do not act at CB1R and the most promising non-psychoactive anticonvulsant phytocannabinoid studied thus far is CBD, which exerts effects via, as yet unknown, non- CB1R mechanisms in vitro, in vivo and in humans (Consroe et al., 1982; Cunha et al., 1980; Jones et al., 2010; Wallace et al., 2001). Because CBD has low affinity for CB1 and CB2 receptors (Pertwee, 2008), CBD may exert its effects through different mechanisms. For instance, it is known that CBD can, at a number of different concentrations in vitro, inhibit adenosine uptake, inhibit FAAH (the enzyme primarily responsible for degradation of the endocannabinoid, anandamide), inhibit anandamide reuptake, act as a TRPA1 receptor agonist, a TRPM8 receptor antagonist, a 5-HT1A receptor agonist, a T-type calcium channel inhibitor and a regulator of intracellular calcium (Izzo et al., 2009).

Here, we have used molecular methods to further investigate the anticonvulsant potential of CBD’s propyl analogue, CBDV (Hill et al., 2012a). Although first isolated in 1969 (Vollner, Bieniek & Korte, 1969), little is known about CBDV’s pharmacological properties (Izzo et al., 2009). Scutt and Williamson reported CBDV to act via CB2 cannabinoid receptor-dependent mechanisms but direct CB2 receptor effects were not shown (Scutt & Williamson, 2007). De Petrocellis reported differential CBDV effects at transient receptor potential (TRP) channels in vitro, noting potent human TRPA1, TRPV1 and TRPV2 agonism and TRPM8 antagonism (De Petrocellis et al., 2011; De Petrocellis et al., 2012). CBDV has also been reported to inhibit diacylglycerol (DAG) lipase-α, the primary synthetic enzyme of the endocannabinoid, 2-arachidonoylglycerol (2-AG) (Bisogno et al., 2003), in vitro (De Petrocellis et al., 2011). However, 2-AG inhibits status epilepticus-like activity in rat hippocampal neuronal cultures (Deshpande et al., 2007a) such that diacylglycerol lipase-α inhibition is unlikely to be anticonvulsant. Furthermore, inhibition of DAG lipase-α by CBDV occurs at high micromolar concentrations (IC50: 16.6 µM) in vitro which are unlikely to have relevance in vivo making it unlikely that CBDV exerts anticonvulsant effects via this route. Although the pharmacological relevance of these effects remains unconfirmed in vivo and the targets identified have not yet been linked to epilepsy, they illustrate an emergent role for multiple, non-CB receptor targets of phytocannabinoids (Hill et al., 2012b; Pertwee, 2010). Furthermore, unlike Δ9-THC, anticonvulsant doses of CBDV exert no detectable effects upon motor function (Hill et al., 2012a) which further supports the assertion that its effects are not CB1R-mediated.

Despite our earlier report showing significant anticonvulsant effects of CBDV in animal models of acute seizure (Hill et al., 2012a), molecular validation of these effects has not yet been undertaken. Here, we evaluated CBDV’s effect (p.o.) on pentylenetetrazole (PTZ)-induced seizures and quantified expression levels of several epilepsy-related genes in tissue from hippocampus, neocortex and prefrontal cortex. Genes of interest were selected on the basis that: (i) their expression was significantly changed in previously published gene expression microarray results from people with epilepsy (PWE) (Helbig et al., 2008; Jamali et al., 2006; van Gassen et al., 2008) and animal models of epilepsy (Elliott, Miles & Lowenstein, 2003; Gorter et al., 2006; Gorter et al., 2007; Okamoto et al., 2010) and (ii) published results (Johnson et al., 2011; Link et al., 1995; McCarthy et al., 1998; Nanda & Mack, 2000; Saffen et al., 1988; Sola, Tusell & Serratosa, 1998; Zhu & Inturrisi, 1993) suggested that expression changes were acute (within a few hours of seizure), making them suitable for study in a model of acute seizure. On this basis, Early growth response 1 (Egr1), Activity-regulated cytoskeleton-associated protein (Arc), Chemokine (C-C motif) ligand 3 (Ccl3), Chemokine (C-C motif) ligand 4 (Ccl4), Brain derived neurotrophic factor (Bdnf), Proenkephalin (Penk) and Neuropeptide Y (Npy) and the downregulated gene, Calcium/calmodulin-dependent protein kinase II alpha (Camk2a) were chosen. FBJ osteosarcoma oncogene (Fos) and Caspase 3 (Casp3) were also selected due to the former’s increased expression in brain regions including hippocampus following experimentally induced seizures (e.g., via PTZ) (Popovici et al., 1990; Saffen et al., 1988) and the latter as a result of increased expression in resected neocortex from people with temporal lobe epilepsy (Henshall et al., 2000).

Material and Methods

Animals

Experiments were conducted in accordance with UK Home Office regulations (Animals (Scientific Procedures) Act, 1986). A total of 51 Wistar-Kyoto rats (Harlan, UK; 3–4 weeks old) were used in this study and ARRIVE guidelines complied with. Animals were group housed in cages of five with water and food supplied ad libitum. Temperature and humidity were maintained at 21°C and 55 ± 10% respectively.

Drug administration

Seizures were induced using PTZ (Sigma, Poole, United Kingdom). After overnight fasting, rats received either vehicle (20% solutol (Sigma) in 0.9%w/v NaCl) or CBDV (400 mg kg−1; GW Pharmaceuticals Ltd., Salisbury, UK) in vehicle by oral gavage. Three and a half hours after vehicle or CBDV administration, rats were challenged (i.p.) with saline or PTZ (95 mg kg−1) and behaviour monitored for 1 h. Animals were euthanised by CO2 overdose and brains immediately removed. Whole hippocampi, neocortices and prefrontal cortices were isolated, snap-frozen in liquid nitrogen and stored at −80°C until RNA extraction.

Analysis of seizure behaviours

Seizure behaviour was video recorded and responses coded exactly as described previously (Hill et al., 2012a). Responses were coded using the following modified Racine seizure severity scale: 0, normal behaviour; 1, isolated myoclonic jerks; 2, atypical clonic seizure; 3, fully developed bilateral forelimb clonus; 3.5, forelimb clonus with tonic component and body twist; 4, tonic–clonic seizure with suppressed tonic phase; 5, fully developed tonic–clonic seizure. Latency to the first sign of seizure was also recorded.

Gene expression analysis

Gene expression was quantified in rat hippocampus, prefrontal cortex and neocortex for four experimental groups: vehicle + saline treated (n = 5), vehicle + PTZ treated (n = 7), CBDV + saline treated (n = 5) and CBDV + PTZ treated (n = 7). Total RNA was extracted using an miRNeasy Mini kit (Qiagen, West Sussex, UK), following the manufacturer’s protocol. RNA purity was assessed spectrometrically at 260/280 nm. RNA integrity was determined by gel electrophoresis. A 28S:18S rRNA ratio of ∼2:1 was taken to indicate intact RNA.

Total RNA (0.5 µg) was reverse-transcribed into cDNA using High Capacity cDNA Reverse Transcription Kits (Applied Biosystems). qPCR assays were carried out in a volume of 14 µl, containing 5 µl cDNA, 2 µl 2.5 µM primer mix (forward and reverse primers) and 7 µl QuantiTect SYBR Green QPCR 2× Master Mix (Qiagen, West Sussex, UK). Samples were processed for 40 cycles on a StepOnePlus™ (Applied Biosystems, Foster City, CA, USA) as follows: denaturation at 95°C for 15 min (one cycle), 40 cycles of denaturation at 95°C for 15 s and annealing at 60°C for 1 min. All samples were analysed in the same plate in a single PCR run and quantification was based on the standard curve method. Standard curves were constructed using cDNA solution diluted fivefold in series for a total of five dilutions and consisted of a mixture of cDNA equally from hippocampus, prefrontal cortex and neocortex of all animals. Sample cDNA concentrations were expressed relative to the concentration of the standard curves. Normalisation of quantitative data was based on a housekeeping gene, β-actin. Values are expressed as a percentage of control (mean of the vehicle + saline group). The following primers were used (parenthesised values are forward and reverse sequence and amplicon length respectively): Ccl3 (5′-TGCCCTTGCTGTTCTTCTCTGC-3′, 5′-TAGGAGAAGCAGCAGGCAGTCG-3′, 96), Ccl4 (5′-CGCCTTCTGCGATTCAGTGC-3′, 5′-AAGGCTGCTGGTCTCATAGTAATCC-3′, 127), Npy (5′-TCGTGTGTTTGGGCATTCTGGC-3′, 5′-TGTAGTGTCGCAGAGCGGAGTAG-3′, 111), Arc (5′-AGGCACTCACGCCTGCTCTTAC-3′, 5′-TCAGCCCCAGCTCAATCAAGTCC-3′, 146), Bdnf (5′-AGCCTCCTCTGCTCTTTCTGCTG-3′, 5′-TATCTGCCGCTGTGACCCACTC-3′, 150), Egr1 (5′-AGCCTTCGCTCACTCCACTATCC-3′, 5′-GCGGCTGGGTTTGATGAGTTGG-3′, 113), Penk (5′-CCAACTCCTCCGACCTGCTGAAAG-3′, 5′-AAGCCCCCATACCTCTTGCTCGTG-3′, 121) and Camk2a (5′-TGAGAGCACCAACACCACCATCG-3′, 5′-TGTCATTCCAGGGTCGCACATCTTC-3′, 142), Fos (5′-TGCGTTGCAGACCGAGATTGC-3′, 5′-AGCCCAGGTCATTGGGGATCTTG-3′, 104), Casp3 (5′-TTGCGCCATGCTGAAACTGTACG-3′, 5′-AAAGTGGCGTCCAGGGAGAAGG-3′,111) and β-Actin (5′-CTCTATCCTGGCCTCACTGTCCACC-3′, 5′-AAACGCAGCTCAGTAACAGTCCGC-3′, 124). Primers were designed using NCBI/Primer-BLAST (http://www.ncbi.nlm.nih.gov/tools/primer-blast/).

Statistics

CBDV effects upon seizure severity and onset latency were assessed by comparing vehicle + PTZ treated and CBDV + PTZ treated groups using a two-tailed Mann-Whitney test and a two-tailed t-test, respectively. Subsequently, animals in the CBDV + PTZ treated group were divided according to median seizure severity score into CBDV ‘responders’ (criterion: seizure severity ≤ median) and ‘non-responders’ (criterion: seizure severity > median) to permit a preliminary subgroup analysis of CBDV effects in these two groups without statistical analysis on subgroups. In qPCR analysis, differences of mRNA expressions between treatment groups were analysed in each brain region using one-way analysis of variance (one-way ANOVA) followed by Tukey’s test. Correlations between seizure severity and mRNA expression in the CBDV + PTZ treated group were analysed using Spearman’s rank correlation coefficient. A preliminary assessment of gene expression changes for CBDV ‘responders’ and ‘non-responders’ was performed, in which differences of mRNA expressions between the vehicle + PTZ treated and the CBDV responder or non-responder subgroups were analysed in each brain region by two-tailed t-test. Since samples from each brain region were analysed on physically separate PCR plates, no comparisons of seizure or drug effects between brain areas were made. Differences were considered statistically significant when the P ≤ 0.05.

Results

Anticonvulsant effects of CBDV on PTZ-induced acute seizures

400 mg kg−1 CBDV significantly decreased seizure severity (vehicle: 5; CBDV: 3.25; P < 0.05) and increased latency to the first seizure sign (vehicle: 60 s; CBDV: 272 s; P < 0.05; Figs. 1A and 1B). Responses of CBDV + PTZ animals sub-grouped into CBDV responders (criterion: seizure severity ≤ 3.25; n = 10) and non-responders (criterion: seizure severity > 3.25; n = 10) showed clear behavioural differences (Figs. 1C and 1D) where CBDV responders exhibit lower seizure severity and increased onset latency.

Figure 1 Anticonvulsant effects of CBDV on PTZ-induced acute seizures.

(A) Plot showing median seizure severity in the vehicle- and CBDV-treated groups following PTZ administration. (B) Plot showing latency (seconds) to the first seizure sign in the vehicle- and CBDV-treated groups. (C) Seizure severity after sub-grouping CBDV treated group animals into CBDV non-responders and CBDV responders. (D) Latency (seconds) to the first seizure sign after subgrouping CBDV treated group animals into CBDV non-responders and CBDV responders. In seizure severity plots, median seizure severity is represented by a thick horizontal line, the 25th and the 75th percentiles are represented by the box and maxima and minima are represented by ‘whiskers’. Latency to the first seizure sign was presented as mean ± SEM. ∗, P < 0.05 by Mann-Whitney Test vs vehicle group; #, P < 0.05 by t-test vs vehicle group.

Effects of PTZ treatment on mRNA expression of epilepsy-related genes in the hippocampus, neocortex and prefrontal cortex

PTZ treatment significantly upregulated Fos mRNA expression in neocortex (P = 0.0001) and prefrontal cortex (P = 0.0003; Table 1) whilst hippocampal Fos mRNA expression only showed a trend to increase (P = 0.1089). Egr1 mRNA expression was significantly upregulated by PTZ treatment in the hippocampus (P = 0.0244), neocortex (P = 0.0001) and prefrontal cortex (P < 0.0001) whilst Arc mRNA expression was also significantly upregulated by PTZ treatment in the hippocampus (P = 0.0374), neocortex (P = 0.0039) and prefrontal cortex (P = 0.0038). Expression of Ccl4 mRNA was significantly upregulated only in the prefrontal cortex (P = 0.0220) by PTZ treatment. Trends toward an increase of Ccl4 mRNA expression in the hippocampus (P = 0.1720) and neocortex (P = 0.1093) by PTZ treatment were seen. Expression of Bdnf mRNA was significantly upregulated in the neocortex (P = 0.0308) and prefrontal cortex (P = 0.0345) but only a trend towards increased expression in the hippocampus was seen (P = 0.0564). mRNA expression of Casp3, Npy, Penk, Ccl3 and Camk2a were not significantly changed by any treatment.

Table 1 Relative mRNA expression levels of epilepsy-related genes in the hippocampus (HIP), neocortex (Nctx) and prefrontal cortex (PFC).

Expressions of Fos, Egr1, Arc, Ccl4 and Bdnf were upregulated by PTZ treatment. mRNA levels are presented as a fold change vs mean level of vehicle + saline treated group (data are expressed as mean ± s.e.m.). Differences between individual groups were assessed by 1-way ANOVA (followed by a Tukey’s post-hoc test if warranted).

Gene official
name	Gene
symbol	GO biological
processes	Brain
region	Vehicle +
Saline	Vehicle +
PTZ	CBDV +
Saline	CBDV +
PTZ	
				Fold change
(N = 5)	Fold change
(N = 7)	Fold change
(N = 5)	Fold change
(N = 7)	
FBJ osteosarcoma oncogene	Fos	Cellular response to calcium ion, cellular response to extracellular stimulus, inflammatory response, nervous system development	HIP	1.0 ± 0.2	55.6 ± 22.2	0.8 ± 0.1	25.4 ± 15.0	
Nctx	1.0 ± 0.3	21.5 ± 3.5 **	0.7 ± 0.1	13.2 ± 2.8*	
PFC	1.0 ± 0.1	20.0 ± 3.8**	0.8 ± 0.1	13.5 ± 2.3*	
Caspase 3	Casp3	Apoptosis, intracellular signal transduction	HIP	1.0 ± 0.1	0.9 ± 0.1	0.9 ± 0.1	0.9 ± 0.1	
Nctx	1.0 ± 0.1	1.1 ± 0.1	1.0 ± 0.1	1.1 ± 0.1	
PFC	1.0 ± 0.0	1.1 ± 0.1	0.9 ± 0.1	0.9 ± 0.1	
Early growth response 1	Egr1	Cellular response to drug, cellular
response to growth factor stimulus,
cellular response to steroid
hormone stimulus, circadian
rhythm, interleukin-1-mediated
signaling pathway	HIP	1.0 ± 0.0	6.1 ± 1.5*	0.8 ± 0.1	3.6 ± 1.1	
Nctx	1.0 ± 0.1	3.0 ± 0.4**	0.7 ± 0.1	2.5 ± 0.2**	
PFC	1.0 ± 0.1	2.7 ± 0.3**	0.8 ± 0.1	2.2 ± 0.2**	
Activity-regulated cytoskeleton-associated
protein	Arc	Regulation of neuronal synaptic plasticity, endocytosis	HIP	1.0 ± 0.1	8.6 ± 2.5*	0.8 ± 0.1	4.2 ± 1.7	
Nctx	1.0 ± 0.2	5.0 ± 1.1**	0.6 ± 0.1	3.4 ± 0.5	
PFC	1.0 ± 0.1	4.4 ± 0.9**	0.7 ± 0.1	3.0 ± 0.4	
Neuropeptide Y	Npy	Feeding behavior, negative
regulation of blood pressure,
synaptic transmission	HIP	1.0 ± 0.1	0.9 ± 0.1	1.0 ± 0.1	1.0 ± 0.1	
Nctx	1.0 ± 0.1	1.0 ± 0.1	1.0 ± 0.1	1.1 ± 0.1	
PFC	1.0 ± 0.1	0.9 ± 0.0	1.0 ± 0.1	0.9 ± 0.0	
Chemokine (C-C motif)
ligand 4	Ccl4	Chemotaxis, inflammatory
response	HIP	1.0 ± 0.1	16.7 ± 5.9	0.7 ± 0.2	7.9 ± 6.3	
Nctx	1.0 ± 0.3	36.0 ± 14.8	1.4 ± 0.3	15.4 ± 8.8	
PFC	1.0 ± 0.2	13.3 ± 3.4*	1.0 ± 0.2	7.9 ± 3.0	
Chemokine (C-C motif)
ligand 3	Ccl3	Chemotaxis, elevation of cytosolic calcium ion concentration, inflammatory response	HIP	1.0 ± 0.2	8.8 ± 3.7	1.1 ± 0.2	5.3 ± 3.7	
Nctx	1.0 ± 0.2	21.1 ± 10.5	1.6 ± 0.2	13.0 ± 6.2	
PFC	1.0 ± 0.1	16.4 ± 6.3	1.5 ± 0.1	13.5 ± 5.9	
Brain derived
neurotrophic factor	Bdnf	Neuron differentiation,
positive regulation of long-term
neuronal synaptic plasticity,
glutamate secretion	HIP	1.0 ± 0.1	2.6 ± 0.6	0.9 ± 0.1	1.7 ± 0.3	
Nctx	1.0 ± 0.0	2.5 ± 0.4*	0.9 ± 0.1	2.1 ± 0.4	
PFC	1.0 ± 0.1	2.1 ± 0.4*	1.1 ± 0.2	1.9 ± 0.2	
Proenkephalin	Penk	Behavioral fear response,
sensory perception of pain	HIP	1.0 ± 0.1	1.2 ± 0.2	1.1 ± 0.1	1.1 ± 0.1	
Nctx	1.0 ± 0.2	1.1 ± 0.2	0.8 ± 0.1	1.1 ± 0.1	
PFC	1.0 ± 0.2	0.9 ± 0.2	1.1 ± 0.2	0.9 ± 0.2	
Calcium/calmodulin-
dependent protein
kinase II alpha	Camk2a	Calcium ion transport, ionotropic
glutamate receptor signaling
pathway, protein phosphorylation,
regulation of neuronal synaptic
plasticity, regulation of
neurotransmitter secretion	HIP	1.0 ± 0.1	0.9 ± 0.0	0.9 ± 0.1	0.9 ± 0.1	
Nctx	1.0 ± 0.1	0.9 ± 0.1	1.0 ± 0.1	1.0 ± 0.1	
PFC	1.0 ± 0.1	1.0 ± 0.1	1.1 ± 0.1	1.0 ± 0.1	
Notes.

* P < 0.05 vs vehicle + saline group.

** P < 0.01.

Effects of CBDV upon PTZ-induced mRNA expression of epilepsy-related genes in the hippocampus, neocortex and prefrontal cortex

Fos and Egr1 mRNA expression were significantly upregulated in the neocortex (P = 0.0201 and P = 0.0033, respectively) and the prefrontal cortex (P = 0.0156 and P = 0.0023, respectively) in the CBDV + PTZ treated group. Although there were no statistically significant changes in the expression levels of any other genes between the vehicle + saline and CBDV + PTZ treated groups which suggests an inhibitory effect of CBDV on PTZ-induced upregulation of gene expression, neither were statistically significant differences in gene expression levels between the vehicle + PTZ and CBDV + PTZ treated groups found. However, when potential correlations between the behavioural measure of seizure severity and mRNA expression levels of Fos, Egr1, Arc, Bdnf and Ccl4 in the CBDV + PTZ treated group were examined using Spearman’s rank correlation coefficient, mRNA expression levels of these genes were highly correlated with seizure severity in the majority of brain regions examined (Fig. 2: hippocampus, Fig. 3: neocortex and Fig. 4: prefrontal cortex). Fos mRNA expression correlated with seizure severity in the hippocampus (R2 = 0.91, P = 0.0008), neocortex (R2 = 0.91, P = 0.0008) and prefrontal cortex (R2 = 0.91, P = 0.0008) of the CBDV + PTZ treated group. Egr1 mRNA expression was correlated with seizure severity only in the hippocampus (R2 = 0.91, P = 0.0008) whilst Arc mRNA expression was correlated with seizure severity in the hippocampus (R2 = 0.91, P = 0.0008), neocortex (R2 = 0.91, P = 0.0008) and prefrontal cortex (R2 = 0.71, P = 0.0175). Bdnf mRNA expression was correlated with seizure severity in the hippocampus (R2 = 0.71, P = 0.0175) and neocortex (R2 = 0.65, P = 0.0291) whilst Ccl4 mRNA expression was correlated with seizure severity in the hippocampus (R2 = 0.91, P = 0.0008), neocortex (R2 = 0.71, P = 0.0175) and prefrontal cortex (R2 = 0.71, P = 0.0175). Together, these suggest a possible contribution of the anti-convulsant effects of CBDV in reduction of mRNA expression of Fos, Egr1, Arc, Bdnf and Ccl4.

Figure 2 Correlation analysis between seizure severity and mRNA expression levels in the hippocampus.

Correlations between mRNA expression of Fos (A), Egr1 (B), Arc (C), Bdnf (D) and Ccl4 (E) and seizure severity were analysed using Spearman’s rank correlation coefficient.

Figure 3 Correlation analysis between seizure severity and mRNA expression levels in the neocortex.

Correlations between mRNA expression of Fos (A), Egr1 (B), Arc (C), Bdnf (D) and Ccl4 (E) and seizure severity were analysed using Spearman’s rank correlation coefficient.

Figure 4 Correlation analysis between seizure severity and mRNA expression levels in the prefrontal cortex.

Correlations between mRNA expression of Fos (A), Egr1 (B), Arc (C), Bdnf (D) and Ccl4 (E) and seizure severity were analysed using Spearman’s rank correlation coefficient.

Effects of CBDV treatment on the PTZ-induced increases of the epilepsy-related genes in CBDV responders

Consistent with differing behavioural patterns observed between CBDV responder and non-responder subgroups, alterations in gene expression were also seen. Importantly, changes in gene expression levels between the vehicle + PTZ and the CBDV responder subgroups were most obvious, with few changes seen in gene expression levels between vehicle + PTZ and the CBDV + PTZ non-responder subgroups. Importantly, PTZ-induced increases in gene expression were most reliably suppressed in the hippocampus of CBDV responders, with less obvious suppression in prefrontal cortex and neocortex. The PTZ-induced increase of Fos mRNA expression in CBDV responders was suppressed in the neocortex (P = 0.0274) and the prefrontal cortex (P = 0.0337), and there was a strong trend towards a decrease in the hippocampus (P = 0.0579; Fig. 5A). The PTZ-induced increase of Egr1 mRNA expression was suppressed in the hippocampus (P = 0.0234) of CBDV responders, but less obviously so in the neocortex (P = 0.1837) and the prefrontal cortex (P = 0.1038; Fig. 5B). The increase in Arc mRNA expression induced by PTZ treatment was also suppressed in the hippocampus (P = 0.0221) of CBDV responders, and there were strong trends towards decreases in the neocortex (P = 0.0643) and the prefrontal cortex (P = 0.0879; Fig. 5C). The increase of Bdnf mRNA expression following PTZ treatment was most suppressed in the hippocampus (P = 0.0441) of CBDV responders whilst less decreases were seen in the neocortex (P = 0.1099) and prefrontal cortex (P = 0.4128; Fig. 5D). Finally the PTZ-induced increase of Ccl4 mRNA expression was suppressed in the hippocampus (P = 0.0323) and the prefrontal cortex (P = 0.0459), and there was a strong trend towards a decrease in the neocortex (P = 0.0942; Fig. 5E). On the other hand, neither statistically significant decreases nor trends towards decreases in the gene expressions were found in the CBDV non-responder subgroup.

Figure 5 Subgroup-analysis of mRNA levels of epilepsy-related genes in CBDV responders and nonresponders.

Subgrouping CBDV + PTZ treated animals into responders (criterion: seizure severity ≤3.25) and non-responders (criterion: seizure severity > 3.25) revealed that the PTZ-induced increases of mRNA expression of Fos (A), Egr1 (B), Arc (C), Bdnf (D) and Ccl4 (E) were significantly suppressed in brain regions examined from the CBDV responder subgroup. mRNA levels are presented as a fold change vs mean level of vehicle + saline treated group (data are expressed as mean ± s.e.m.). ∗, P < 0.05 by t-test vs vehicle + PTZ group.

Discussion

PTZ treatment upregulated (significant increase or statistically strong trend to increase) mRNA expression coding for Fos, Egr1, Arc, Ccl4 and Bdnf in all brain regions tested. Clear correlations between seizure severity and mRNA expression were observed for these genes in the majority of brain regions of CBDV + PTZ treated animals and mRNA expression of these genes was suppressed in the majority of brain regions examined from the CBDV responder subgroup. Upregulation of Fos and Egr1 mRNA expression following PTZ treatment has previously been reported in rat hippocampi (Saffen et al., 1988) and both Fos and Egr1 are transcription factors belonging to IEG (immediate early gene) family which is transiently and rapidly activated following a variety of cellular stimuli. IEGs can identify activated neurons and brain circuits since seizure activity, and other excitatory stimuli, can induce rapid and transient Fos expression increases (Herrera & Robertson, 1996), making it a useful metabolic marker for brain activity (Dragunow & Faull, 1989). Fos expression level in the brain is typically low under basal conditions and is induced in response to extracellular signals such as ions, neurotransmitters, growth factors and drugs and is closely linked to the induction of transcription of other genes (Kovacs, 2008). Fos induction also correlates with the mossy fibre sprouting (Kiessling & Gass, 1993; Popovici et al., 1990) that occurs during epileptogenesis and may play a role in the subsequent manifestation of seizure symptoms. Like Fos, Egr1 also activates transcription of other genes (Beckmann et al., 1997; Christy & Nathans, 1989) and is considered to play an important role in neuronal plasticity (Knapska & Kaczmarek, 2004). Furthermore, the expression of Fos and Egr1 in seizure onset regions in PWE strongly correlates with interictal spiking (Rakhade et al., 2007). Thus, suppression of Fos and Egr1 mRNA expression are consistent with ameliorative drug effects on seizures, epileptogenesis and/or epilepsy. In addition, increased Arc mRNA expression in rat hippocampus (0.5–4 h) and cortex (0.5–1 h) after PTZ treatment has also been reported (Link et al., 1995). It has been reported that newly synthesised Arc mRNA is selectively localised in active dendritic segments and that Arc plays a role in activity-dependent plasticity of dendrites (Lyford et al., 1995; Steward et al., 1998). Arc is induced by hippocampal seizures, and glutamatergic neurons increase Arc expression in response to increased synaptic activity (Korb & Finkbeiner, 2011), implying a relationship between seizure activity and Arc expression. Ccl4 is a proinflammatory chemokine that is known as a chemo-attractant for monocytes and T cells and has been suggested to play a part in various nervous system pathologies such as inflammation, trauma, ischemia and multiple sclerosis (Semple, Kossmann & Morganti-Kossmann, 2010). Although a relationship between CCL4 and epilepsy is unclear, a relationship between epilepsy and immune response has been suggested (Vezzani & Granata, 2005). Moreover, increased Ccl4 mRNA expression has been reported in rat hippocampi and temporal lobe tissue following status epilepticus events triggered by electrical stimulation of the amygdala (Guzik-Kornacka et al., 2011). In the present study, PTZ-induced increase of Ccl4 expression was suppressed in CBDV responders, although whether this is a direct anti-inflammatory effect of CBDV or an indirect effect of reduced seizure severity remains unknown. Increased expression of mRNA coding for Bdnf was confirmed in rat hippocampus after PTZ treatment (Nanda & Mack, 2000). BDNF is one of many neurotrophic factors and is known to promote survival and growth of a variety of neurons in addition to strengthening excitatory (glutamatergic) synapses (Binder & Scharfman, 2004). BDNF is involved in the control of hippocampal plasticity and is thought to play an important role in epileptogenesis and in temporal lobe epilepsy (Binder et al., 2001; Scharfman, 2002), suggesting therapeutic importance for control of Bdnf expression.

Conclusions

We have confirmed upregulation of mRNA expression coding for Fos, Egr1, Arc, Ccl4 and Bdnf in the brains of rats treated with PTZ and shown that PTZ-induced increases of mRNA expression for these genes were suppressed in CBDV responders, and not animals that failed to respond to CBDV treatment. Overall, we provide molecular evidence that directly supports behavioural evidence that CBDV exerts significant anticonvulsant effects via oral and other routes of administration (Hill et al., 2012a). Whether gene expression changes demonstrated here also underlie cellular and molecular mechanisms by which CBDV exerts its anticonvulsant effect presently remains unknown. However, these results provide important acute biomarkers for additional investigation in models of the progressive disorder and following longer term CBDV treatment.

We thank GW Pharmaceuticals Plc. for providing CBDV.

Additional Information and Declarations

Competing Interests

Author Contributions

Animal Ethics

Data Deposition

GW Pharmaceuticals Plc. and Otsuka Pharmaceutical Co, Ltd. are collaborators on an epilepsy research project. As an employee of Otsuka Pharmaceutical Co, Ltd., Naoki Amada is participating in this collaborative project, and Naoki Amada’s PhD project is funded by this research collaboration. GW Pharmaceuticals Plc. provided materials (CBDV). Yuki Yamasaki is an employee of Otsuka Pharmaceutical Co, Ltd., and holds stocks in the company. Benjamin Whalley is an Academic Editor for PeerJ.

Naoki Amada and Yuki Yamasaki conceived and designed the experiments, performed the experiments, analyzed the data, wrote the paper.

Claire M. Williams and Benjamin J. Whalley conceived and designed the experiments, wrote the paper.

The following information was supplied relating to ethical approvals (i.e., approving body and any reference numbers):

All work involving the use of animals was reviewed by the University of Reading Local Ethical Review Panel in addition to being conducted under the authority of UK Home Office Project Licence 30/2538 issued to Dr Benjamin Whalley under the Animals (Sceintific Procedures) Act, 1986.

The following information was supplied regarding the deposition of related data:

CentAUR (Central Archive at the University of Reading)

http://centaur.reading.ac.uk/.

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
