# Peer review of "Cannabidivarin (CBDV) suppresses pentylenetetrazole (PTZ)-induced increases in epilepsy-related gene expression"

_PeerJ, doi:10.7717/peerj.214_

## Round 0.1 · original submission · Major Revisions

The paper mainly describes an association between seizure and gene expression changes. Majority of data was already reported. The study would benefit from investigations on protective role of CBVD on seizures.

Reviewer 1 ·

Basic reporting

Figure 1 legend has spaces between first letter and rest of word for 'm edian' l atency' and 'v ehicle'.

Figure 1 legend for A and B is very terse. It might be better to indicate 'plot showing latency to seizure' or similar.

Units in Figure 1 A and B are not indicated on the axes e.g. seizure latency should be in seconds (s).

Units of Figures 1 and 2 axis labels differ in capitalisation Seizure Severity and Seizure severity.

Figure 3 needs * over the significant bars.

conclusion lines 277/8 - no routes other than oral route were tested so why include this mention of other routes?

Experimental design

The experimental design and statistics were of high standard. Was a two-tailed test strictly necessary given that expected a priori assumption would be for a unidirectional change towards lower severity and longer latency?

Validity of the findings

soundly analysed (too conservatively if anything).

Additional comments

this was a well-preented and written manuscript requiring only minor modifications to make it publishable and represents some fascinating data regarding these drugs and their potential in epilepsy.

Reviewer 2 ·

Basic reporting

In the first part of the introduction authors first described how agonists of CB1R are anticonvulsant and antiepileptiform drugs. Subsequently, describing the anticonvulsant effect of Cannabidivarin (CBDV) they suggest a probable CB1R-independent effect, that also include a possible reduction in the CB1R agonist 2-arachidonoylglycerol (2-AG). Authors should better explain these differences otherwise readers can be confused about the activity of cannabinoids on seizures and epilepsy.

The description of the seizures severity in methods section should be better explained and not only referenced.

Experimental design

Experiments are not well conducted for the purpose of the paper. Authors wanted to clarify the molecular mechanism involved in CBDV-mediated seizure suppression. With the experiments performed in this paper they only show that seizures induce an upregulation of several genes (already know), and that suppressing seiures with CBVD can partially reduce the seizures-induced upregulation. There is not molecular mechanism about the effect of CBVD. The effect on genes expression is likely to be mediated by the decrease seizure severity induced by CBVD. Moreover, the effect on seizure onset lead to the hypothesis that the effect of CBVD is very fast and not compatible with the time required for the genes expression.

Validity of the findings

the majority if not all the data reported on this paper were already known. Authors wanted to investigate the CBVD-mediated mechanism responsible for the reduction in seizure activity. The experiments performed are not appropriated since they only reported that a reduction in seizure activity is associated with a reduction of expression of several genes. This is probably a consequence of the reduction of seizure activity and thus is not related with the molecular mechanism induced by CBVD.
The purpose of the paper is not reached and the experiments performed are incorrect.

Additional comments

Authors should re-think all the experimental plan, since their experiments are not appropriated with the purpose of the paper.

---

## Round 0.2 · Major Revisions

The major problem with this paper is the lack of any insight on the molecular mechanism underlying the effect of CBVD. This evidence could be of master importance to dissect the effect of the drug from that of decreased seizures activity in epilepsy. The Authors draw a conclusion based on changes in mRNA expression that could be misleading for the scientific community. The Authors should try to address the molecular mechanism with few experiments and provide an hypothesis for the effect of CBVD at molecular level.

Reviewer 1 ·

Basic reporting

No any comment about this point.

Experimental design

After modifications and explanations added after the first submission, still the principal research question is not answered. Specifically all the results presented by the authors can likely be a consequence of reduced seizures and not an effect mediated by CBDV.

Validity of the findings

The finding, still seems not appropriate and misunderstaning on the role of cannabinoid system in epilepsy can originate.

Additional comments

The paper still do not represent an advancement in understanding the role of cannabinoid system in epilepsy. Lack of novelty and lack of clarification of the role of CBVD in epilepsy are the major problems of this paper. Still all the results presented by the authors can be addressed to a decreased seizures activity, thus the possible effect of CBVD is not clarified.

---

## Round 0.3 · accepted · Accept

The present manuscript is acceptable for publication.